# The Impact of Pre-Chemotherapy Body Composition and Immunonutritional Markers on Chemotherapy Adherence in Stage III Colorectal Cancer Patients

**DOI:** 10.3390/jcm12041423

**Published:** 2023-02-10

**Authors:** Soohyeon Lee, Dong Hyun Kang, Tae Sung Ahn, Seung Soo Kim, Jong Hyuk Yun, Hyun Jung Kim, Seoung Hee Seo, Tae Wan Kim, Hye Jeong Kong, Moo Jun Baek

**Affiliations:** 1Department of Surgery, Soonchunhyang University College of Medicine, Soonchunhyang University Cheonan Hospital, Cheonan 31151, Republic of Korea; 2Department of Radiology, Soonchunhyang University College of Medicine, Soonchunhyang University Cheonan Hospital, Cheonan 31151, Republic of Korea; 3Department of Medical Life Science, Soonchunhyang University, Asan 31538, Republic of Korea

**Keywords:** body composition, chemotherapy adherence, relative dose intensity, colorectal cancer

## Abstract

Patients with colorectal cancer (CRC) often fail to complete full-course chemotherapy with a standard dose due to various reasons. This study aimed to determine whether body composition affects chemotherapy adherence in patients with CRC. The medical records of 107 patients with stage III CRC who underwent adjuvant folinic acid, fluorouracil and oxaliplatin (FOLFOX) chemotherapy at a single center between 2014 and 2018 were analyzed retrospectively. Blood test results for selected immunonutritional markers were analyzed and body composition was measured through computed tomography. Univariate and multivariate analyses were performed on low and high relative dose intensity (RDI) groups, based on an RDI of 0.85. In the univariate analysis, a higher skeletal muscle index was correlated with a higher RDI (*p* = 0.020). Psoas muscle index was also higher in patients with high RDI than in those with low RDI (*p* = 0.026). Fat indices were independent of RDI. Multivariate analysis was performed for the aforementioned factors and results showed that age (*p* = 0.028), white blood cell count (*p* = 0.024), and skeletal muscle index (*p* = 0.025) affected RDI. In patients with stage III CRC treated with adjuvant FOLFOX chemotherapy, a decrease in RDI was related to age, white blood cell count, and skeletal muscle index. Therefore, if we adjust the drug dosage in consideration of these factors, we can expect an increased treatment efficiency in patients by increasing chemotherapy compliance.

## 1. Introduction

According to the American Cancer Society, colorectal cancer (CRC) is the third most common cause of cancer-related deaths worldwide with an incidence of approximately 4% in both men and women [1]. To reduce mortality from CRC, colonoscopy and imaging modalities for early detection have been developed, and treatment guidelines, including radiation therapy and various anticancer drugs, have been presented, in addition to surgery [2]. Recently, drugs used according to molecular biomarkers, which differ for each patient, have been actively applied, including immunotherapy and targeted therapy [3]. Approximately 48.5% of patients with CRC are diagnosed at stage III/IV of the disease, but with the help of the aforementioned efforts, the 5-year overall survival rate has increased to 74.3%. Patients diagnosed with stage III disease can receive first-line treatments, folinic acid, fluorouracil and oxaliplatin (FOLFOX) or capecitabine and oxaliplatin (CAPEOX), which have a disease-free survival of 63.3% among stage III patients with CRC treated by aggressive chemotherapy [4].

However, in many CRC cases, adjuvant chemotherapy cannot be initiated or fully completed due to post-operative complications, worsening of underlying conditions, or other reasons [5]. The search for a solution to the causes of these problems has highlighted the significance of malnutrition and immunonutrition. Malnutrition is common in patients with cancer. It can impair immune function and increase the risk of infection. Malnutrition has been shown to increase the risk of morbidity and mortality, as well as affect treatment intolerance in patients with cancer [6,7]. Chemotherapy can lead to mucositis, enteritis, or colitis due to epithelial cell damage; these side effects may exacerbate anorexia, thereby worsening existing malnutrition. Chemotherapy induces inflammation and immunological suppression, leading to toxicity [8]. Therefore, identification and supplementation of immunonutritional status can modulate the inflammatory and immune response in patients with cancer and reduce acute toxicity [9]. Body surface area (BSA) and body mass index (BMI) are used to determine the appropriate dose of anticancer drugs. However, a persistent problem is that BSA does not fully represent lean body mass (LBM), which refers to non-adipose tissue mass. BSA does not reflect the differences in individual responses to drugs [10,11]. Other studies have reported that BMI is also unrelated to chemotherapy adherence. Therefore, if the dose of anticancer drugs is determined by BMI or BSA, it does not prevent excess toxicity associated with low LBM [12,13]. According to several studies, body composition is an accurate indicator for evaluating nutritional status, and body composition, including muscle and fat mass, can be easily calculated using computed tomography (CT) [14]. Therefore, several studies have assessed the effects of body composition or immunonutrition on the survival of patients with cancer. However, studies that consider both the effects of body composition and immunonutrition on patients undergoing chemotherapy are insufficient.

We investigated the impact of nutrition and body composition on chemotherapy adherence and compliance in patients with stage III CRC receiving adjuvant FOLFOX chemotherapy as a first-line treatment and further considered indicators to increase patient compliance with chemotherapy before initiation of other chemotherapy.

## 2. Materials and Methods

### 2.1. Study Population

This retrospective study used the medical records of patients with CRC who underwent surgery at Soonchunhyang University, Cheonan Hospital, Cheonan-si, South Korea between January 1, 2014, and July 31, 2018. Patients who were diagnosed with stage III CRC based on the 7th edition Cancer Staging Manual of the American Joint Committee on Cancer and received adjuvant FOLFOX chemotherapy as a first-line treatment were considered for this study. Ethics approval was obtained from the Institutional Review Board of Soonchunhyang University, Cheonan Hospital. A total of 121 patients were selected. The exclusion criteria were as follows: loss of medical records, underlying diseases, such as liver cirrhosis and psychiatric disorders that required medication, the presence of sarcopenia prior to surgery due to intestinal obstruction, and cancer of other organs diagnosed within the last two years. Ultimately, 107 patients were enrolled in our study, and baseline characteristics, including tumor markers, were described.

### 2.2. Chemotherapy Adherence and Outcomes

Chemotherapy was initiated within 6–8 weeks after surgery, and FOLFOX-4 regimen was administered as follows: oxaliplatin 85 mg/m^2^ on day 1, leucovorin 200 mg/m^2^/day on days 1 and 2, and 5-fluorouracil (5-FU) bolus 400 mg/m^2^/day, followed by the continuous infusion of 600 mg/m^2^/day on days 1 and 2. The regimen was planned to be repeated every two weeks and implemented for a total of 12 cycles. Early discontinuation was defined as cases where a patient received less than six cycles of chemotherapy. The relative dose intensity (RDI) was used to evaluate chemotherapy adherence. The RDI was calculated as the actual dose intensity/projected dose intensity for each drug separately in the treatment regimen; the total RDI was calculated as the average of the RDI values of all medications. The dose intensity was expressed as total drug (mg))/body surface area x number of weeks. The RDI was divided into two categories: high (≥0.85) and low (<0.85) RDI [12]. In this study, the primary outcome was whether RDI was affected by the patient’s nutritional status and body composition measured prior to the initiation of chemotherapy.

### 2.3. Markers and Parameters of Nutrition and Inflammation-Related CRC

We confirmed the results of the blood test performed before initiation of the first chemotherapy session to evaluate the nutritional status of patients. A total of nine markers and parameters (total protein (g/dL), serum albumin (g/dL), total cholesterol (mg/dL), hemoglobin (g/dL), white blood cell count (WBC) (10^3^/μL), controlling nutritional status (CONUT) score (point), prognostic nutritional index (PNI), neutrophil–lymphocyte ratio (NLR), and platelet–lymphocyte ratio (PLR)) were used to compare the difference between the low and high RDI groups. The CONUT score is a screening tool to identify undernourished patients [15]; it is the sum of scores of three markers: serum albumin (g/dL), total lymphocyte count (/mm^3^), and total cholesterol (mg/dL). The PNI was calculated as [10 × serum albumin (g/dL) + 0.005 × total lymphocyte count (cells/mm^3^)]; it has been used as an independent prognostic indicator of cancer in many studies [16]. Additionally, the NLR and PLR are measurements of systemic inflammation in CRC, and pretreatment levels of these nutrition–inflammation biomarkers (NIBs) are associated with prognosis and treatment outcomes.

### 2.4. Measures of Body Composition

Non-contrast-enhanced CT data were obtained from pre-therapeutic abdominopelvic CT studies (GoldSeal CT750, GE Healthcare, Chicago, IL, USA) and a communicating system (PACS) before surgery. Semi-automated segmentation was performed using 3DSlicer Software (version 5.0.2, www.slicer.org, accessed on 28 June 2022). The region of interest (ROI) of skeletal muscles, and visceral and subcutaneous fat were on a single axial slice of the CT scan at the center of the third lumbar (L3) vertebra level, distinguishing muscle from visceral and subcutaneous fat and skeletal muscles using anatomical knowledge and standard Hounsfield units (HUs). The L3 skeletal muscles included the psoas, lumbar, erector spinae, transversus abdominis, internal and external oblique, and rectus abdominis muscles. According to various studies, [17] we considered a threshold of −190 to −30 HU as subcutaneous fat, −150 to −50 HU as visceral fat, and −29 to 150 HU as skeletal muscle. The total fat area (TFA) was determined as the sum of the subcutaneous fat area (SFA) and visceral fat area (VFA). To calculate the index (cm^2^/m^2^) for subcutaneous fat (SFI), visceral fat (VFI), total fat (TFI), skeletal muscle (SMI), and psoas muscle (PMI), cross-sectional areas of fat were adjusted for patient height. A visceral–subcutaneous fat ratio used to assess abdominal fat distribution is calculated by dividing VFA by SFA.

### 2.5. Statistical Analysis

To calculate descriptive statistics, we used statistical package for the social sciences (SPSS) statistics ver. 26, and two-sided *p*-values less than 0.05 were considered statistically significant. Baseline characteristics are reported as medians (range) or numbers (percentage), and two groups according to RDI less than 0.85 were compared in a total of 14 factors including tumor markers. Continuous variables were analyzed using the independent Student’s t-test or the Mann–Whitney test after performing a normality test, and categorical variables were analyzed using a chi-squared test. Univariate regression analysis was performed to analyze the relationship between each parameter/marker and RDI. Multivariate analysis adjusted for age and sex was performed using multivariate logistic regression. Based on the predictive variables obtained by multivariate logistic regression, a nomogram and receiver operating characteristic (ROC) curves were constructed using R version 3.6.3.

## 3. Results

Among the patients with stage III CRC, 107 patients received adjuvant FOLFOX-4 chemotherapy as a first-line treatment. We compared the low and high RDI groups according to an RDI of 0.85 (Table 1). We evaluated the reasons for discontinuing chemotherapy or dose reduction in the two RDI groups based on common terminology criteria for adverse events version 5. In the low RDI group (Appendix A), eight patients with peripheral sensory neuropathy (most frequently occurring reason) and four patients with grade 2 and grade 3 symptoms were included; however, there were no patients with grade 4 symptoms. The next most frequent reason was a decreased neutrophil count, with a total of six patients, including three patients each with grade 3 and grade 4 symptoms, followed by four patients with grade 2 nausea. Other causes were fatigue in two patients, grade 4 lung infection in one patient, and grade 3 abdominal pain in one patient. In the high RDI group, 32 out of 85 patients complained of chemotherapy side effects, and unlike those in the low RDI group, the causes varied (Appendix A). Similar to the low RDI group, nausea, peripheral sensory neuropathy, and decreased neutrophil count were the most common reasons; however, the toxicity grade was generally lower than that of the low RDI group. In addition, urticaria, diarrhea, syncope, and intestinal stoma site bleeding were also causes of dose reduction. If the BMI measured before chemotherapy was >25.0, obesity was considered based on the Asia–Pacific classification of BMI. The distribution by sex was even, and there was no difference between the two groups according to BMI (*p* = 0.949). There were relatively more patients with colon cancer of the left side in the high RDI group, therefore, anterior resection was most commonly performed at 43.5%. However, there was no significant difference between tumor location and surgical method according to RDI. In addition, a stoma was created during surgery in 18.2% of patients with low RDI and 11.8% of patients with high RDI; the difference between the two groups was not significant (*p* = 0.431). Of these, only one patient underwent Hartmann’s procedure: this was the patient with colostomy; thirteen patients underwent loop ileostomy. However, age (odds ratio (OR): 4.010, 95% confidence interval (CI): 1.356–11.863, *p* = 0.009) was found to be a factor affecting RDI. In a histopathology report, the high-risk of stage III CRC was 36.4% in patients with low RDI and 37.6% in those with high RDI (*p* = 0.912). Although statistically insignificant, venous invasion was different between the low and high RDI groups (*p* = 0.052).

Figure 1 is the Lumbar 3 (L3) level abdomino-pelvic CT images of two patients with the same BMI of 30 kg/m^2^. Each part of the body composition was calculated using 3DSlicer Software, and the degree of muscle and fat distribution was remarkably different. That is, it can be seen that estimating the lean body mass and specific body composition using methods such as BMI or BSA is less accurate and has limitations.

The CONUT score was classified into two categories: normal (<2 points) and high (≥2 points). The comparisons of the two groups using univariate analysis showed no difference in CONUT score and PNI when divided by RDI. However, the serum albumin, total protein and WBC count were statistically significant and showed higher values in the high RDI group compared to the low RDI group when t-test or chi-squared tests were additionally performed (Table 2 and Table 3). The NLR and PLR, affected by inflammatory changes in CRC, were not correlated with RDI. The fat ratio considering height was not related to the RDI in both subcutaneous and visceral fat while the muscle index showed a relationship. Furthermore, the SMI and PMI had higher values in the high RDI than in the low RDI group.

As shown in Figure 2, there is a reference line at the top with scores ranging from 0 to 100 points. The scale bars under the reference line represent the effect size of each predictive variable, and the sum of points from each variable and the corresponding predicted probability of low RDI using the two bottom lines can be determined. WBC count = 3 × 10^3^/µL had the highest score (100 points), followed by SMI = 20 cm^2^/m^2^ (87 points), and age < 65 (30 points). The ROC curve analysis of the nomogram that predicted the probability of low RDI showed an AUC value of 0.780 (cut-off point = 0.231) (Figure 3). During the follow-up period, 11 patients with high RDI and 2 patients with low RDI died. The five-year survival rate was 83.07% for patients with high RDI and 90.44% for patients with low RDI (*p* = 0.717). A total of 26 patients experienced relapse, and liver metastasis was the most frequent occurrence. In terms of disease-free survival rate, high RDI was 78.47% and low RDI was 72.06% (*p* = 0.099). Notably, high RDI did not have a positive impact on the survival rate (Appendix A).

## 4. Discussion

Although some studies have been conducted on chemotherapy compliance, body composition, and immunonutritional factors, it is remarkable that our study considered these factors simultaneously to evaluate chemotherapy adherence in stage III CRC for the first time. We found that older the age, lower the WBC count or SMI, and lower the chemotherapy adherence, such as discontinuation of chemotherapy or a reduction in doses. However, a patient’s fat index was statistically insignificant in the relationship with chemotherapy adherence. In addition, the CONUT score and PNI, which are mainly used as nutritional indices, were not related. The nomogram we developed is accurate and reliable, as demonstrated by the AUC value exceeding 0.7 on the ROC curve. This translates that the points a patient accumulates based on known age, WBC count, and SMI values, the higher the possibility of having low RDI by summing up the points. This information can be used to predict patient compliance with chemotherapy before initiating adjuvant chemotherapy.

Skeletal muscle, which accounts for 40% of the body mass, secretes a variety of myokine peptides that affect inflammation, immune function, adipose tissue oxidation, and whole-body metabolism [18,19]. Low muscle mass increases the hospitalization period of patients by increasing post-operative complications, poor physical function, and lower quality of life, and is associated with poor survival [20,21]. In addition, some reports have found that muscle wasting occurs during chemotherapy and has been revealed that a loss of muscle mass and strength is characterized by marked depletion in muscle mitochondrial content, reactive oxygen species (ROS) release, and oxidative to glycolytic muscle fiber shifts [22,23]. The biggest problem affecting adherence and compliance with chemotherapy is the toxicity of chemotherapy. As previously stated, due to muscle loss during chemotherapy, the capacity for metabolism and clearance of drugs is reduced and chemotherapy-induced toxicities are increased [24]. Leukopenia is commonly seen as a chemotherapy toxicity, and the relationship between the inflammatory signaling of intrinsic skeletal muscle and leukopenia has been well established in several studies. To reduce this problem, some studies have emphasized the need for nutritional support during chemotherapy [25] because nutritional intervention improves body weight and muscle mass. However, the detailed mechanism for the relationship between muscle wasting and chemotherapy toxicity is not yet known. Oxaliplatin, which is a component of the FOLFOX regimen, causes a lean tissue mass reduction of approximately 15%, irrespective of food consumption or energy expenditure due to the response of mitochondria in the muscles [26]. 5-fluorouracil (5-FU), which is widely used clinically, reduces skeletal muscle-activated monocytes and macrophages. As a result, weight loss occurs in patients receiving 5-FU, along with general weakness and fatigue [27]. Considering this mechanism, it seems possible to explain the relation of SMI, which is measured before the start of FOLFOX chemotherapy, to chemotherapy adherence.

Unlike previous studies, we investigated whether body composition measured before the initiation of chemotherapy affected adherence to chemotherapy. In our univariate analysis of RDI, both SMI and PMI were statistically significant. In pre-chemotherapy patients with ovarian adenocarcinoma, lower amounts of subcutaneous adipose tissue and skeletal muscle radiodensity were associated with a greater occurrence of grade 3 or higher adverse events, and low skeletal muscle radiodensity increased the risk of death in 3 years [28]. In our study, SMI was still statistically significant in the multiple regression analysis; however, PMI was not significant after adjusting for variables. Contrary to our results, Jung et al. [29] classified sarcopenia based on the psoas muscle cross-sectional area and reported that grades 3–4 toxicity increased, and poor prognosis was observed in patients with sarcopenia. PMI was closely related to hematocrit, hemoglobin, and serum albumin [30]; however, no significant association of RDI was found with these indicators, except serum albumin. In another study, the difference in skeletal muscle mass measured before neoadjuvant chemotherapy in patients with esophageal cancer was not related to the incidence of chemotherapy-related toxicities [31]. Visceral adipose and subcutaneous adipose tissues have different anatomical locations; differences in endocrine function, adipokine secretion, and lipolytic activity also exist [32]. Considering these characteristics, we compared visceral and subcutaneous fat separately, but neither had an effect on RDI. However, Elizabeth et al. [33] reported that the higher the visceral or intramuscular adiposity, the lower the RDI. Justin et al. [13] found that excess visceral and intramuscular adiposity were risk factors for premature discontinuation; however, subcutaneous fat was statistically meaningless. However, other studies on survival confirmed that visceral fat is not related to survival [34], and the more subcutaneous adipose tissue that produces leptin, the more number of patients who can be protected from insulin resistance, which causes cachexia, tumor progression, and metastasis [35]. On this basis, the relationship between adipose tissue and chemotherapy in cancer patients differed across studies. Therefore, additional studies are needed.

Age was also shown to affect RDI, in addition to body composition. Our study data showed that the amount of muscle decreased with increasing age, and muscle loss during chemotherapy in old age was more prominent than that in other age groups. In addition, although the incidence of severe toxicity in patients undergoing chemotherapy does not differ according to age, the older patients often require inpatient treatment for this toxicity, which may reduce chemotherapy compliance [36]. The odds ratio of WBC was 0.671 in the multivariate analysis; therefore, the higher the WBC count, the more chemotherapy compliance in patients. When neutrophils and lymphocytes, which account for the majority of WBCs, were compared separately, lymphocytes were not significantly associated with RDI (OR: 1.215, 95% CI: 0.951–1.553, *p* = 0.119), but neutrophil count was found to be meaningful (OR: 0.624, 95% CI: 0.408–0.955, *p* = 0.030). Considering these results, it was estimated that the more stable the WBC count measured before the initiation of chemotherapy, the lower the risk of neutropenia.

We also found that inflammatory and nutritional indexes, such as CONUT score, PNI, and NLR, might be related to survival rate, but not to RDI. We also investigated whether there was a significant correlation between these parameters and the body composition. PNI was not related to the muscle index but showed a statistically significant positive correlation with VFI (R^2^ = 0.051, t = 2.374, *p* = 0.019). No other meaningful results were obtained. In the C-SCANS study [37], the interaction between SMI and NLR, which is represented by the inflammatory index in multivariable analysis, was not significant.

Previous studies have shown associations between body composition and survival, and those between nutrition/inflammatory factors and survival separately; however, we simultaneously analyzed whether these factors affected chemotherapy adherence in CRC and confirmed this through multiple regression analysis. Although our study was remarkable, it has limitations. First, the sample size was small—the single-center nature of the study could not eradicate selection bias. The index of body composition measured by CT was obtained; however, the actual strength or performance of the muscle was not confirmed because of the retrospective nature of the study. In addition, it did not confirm whether the change in body composition in patients with a low RDI was more pronounced than that in patients with a high RDI on CT after chemotherapy.

## 5. Conclusions

A decrease in RDI was related to age, WBC, and SMI. Therefore, in patients with stage III CRC who received FOLFOX adjuvant chemotherapy, it seems necessary to carefully determine the dose of anti-cancer drugs considering these factors as well as BSA and BMI. As SMI can be measured through abdominal pelvic CT, which is commonly used in CRC staging work-up, it is expected to be useful in clinical practice. In addition, in our study, the frequency and grade of chemotherapy toxicity tended to increase in patients with decreased chemotherapy adherence, and this can be attributed to the increasing compliance by performing chemotherapy at an appropriate dose which will further affect quality of life. In future, it is expected that chemotherapy adherence can be increased by increasing the skeletal muscle mass, which is an independent factor, through post-operative exercise or rehabilitation treatment. We aim to conduct a prospective study with a larger patient population to further examine the relationship between body composition and RDI, considering muscle strength and performance, to enhance the accuracy and details of the nomogram.

## Figures and Tables

**Figure 1 jcm-12-01423-f001:**
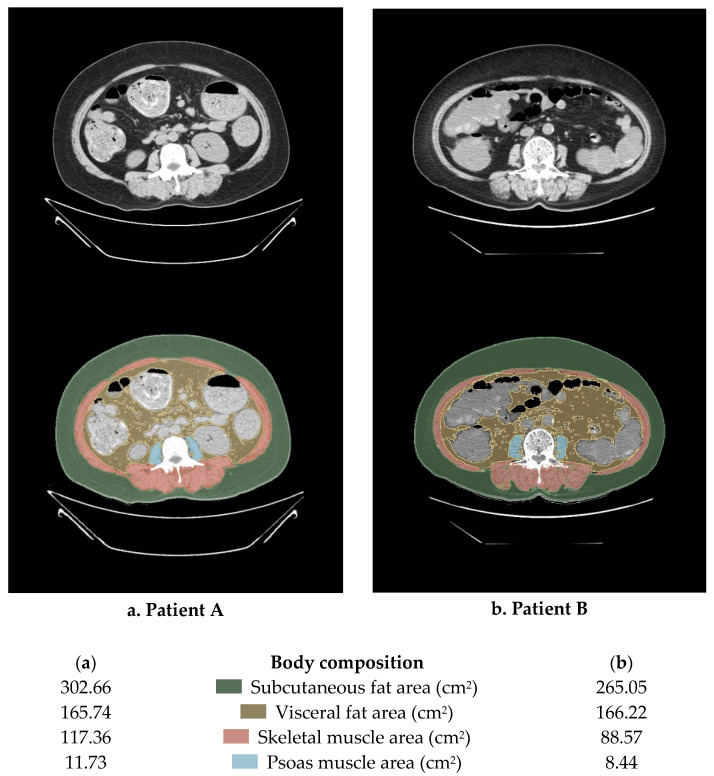
An example of the method for calculating body composition using abdominopelvic computed tomography (APCT). This image is a cut of the L3 and the body mass index of the two patients is the same at 30 kg/m^2^.

**Figure 2 jcm-12-01423-f002:**
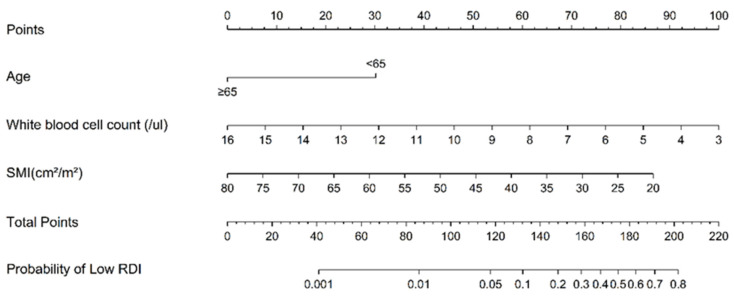
Nomogram used to estimate the probability of low relative dose intensity (RDI) in stage III CRC with adjuvant FOLFOX chemotherapy. SMI, skeletal muscle index.

**Figure 3 jcm-12-01423-f003:**
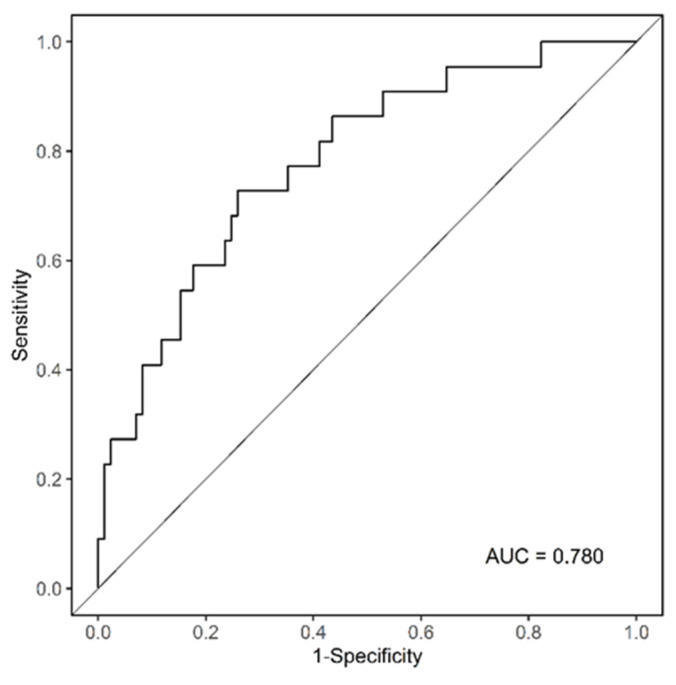
Analysis of the ROC curve to prove the reliability of the nomogram.

**Table 1 jcm-12-01423-t001:** Baseline characteristics of patients with stage III CRC who underwent FOLFOX chemotherapy.

Characteristics		Overall (*n* = 107)	Low RDI (*n* = 22)	High RDI (*n* = 85)	*p*-Value	OR (95% CI)
		(No (%))	(No (%))	(No (%))		
Age	≥65	56 (52.3)	17 (77.3)	39 (45.9)	0.009	4.010 (1.356–11.863)
	<65	51 (47.7)	5 (22.7)	46 (54.1)		
Sex	Male	53 (49.5)	10 (45.5)	43 (50.6)	0.668	0.814 (0.318–2.085)
	Female	54 (50.5)	12 (54.5)	42 (49.4)		
DM	Yes	31 (29.0)	7 (31.8)	24 (28.2)	0.741	1.186 (0.430–3.269)
	No	76 (71.0)	15 (68.2)	61 (71.8)		
HTN	Yes	43 (40.2)	8 (36.4)	35 (41.2)	0.682	0.816 (0.309–2.154)
	No	64 (59.8)	14 (63.6)	50 (58.8)		
Pre-chemotherapy BMI (kg/m^2^)	Underweight (<18.5)	8 (7.5)	2 (9.1)	6 (7.1)	0.949	-
	Normal (18.5–24.9)	74 (69.2)	15 (68.2)	59 (69.4)		
	Obese (≥25.0)	25 (23.4)	5 (22.7)	20 (23.5)		
Tumor location ^1^	Right side	34 (31.8)	9 (40.9)	25 (29.4)	0.302	0.602 (0.228–1.587)
	Left side	73 (68.2)	13 (59.1)	60 (70.6)		
Name of operation	Right hemicolectomy	32 (29.9)	9 (40.9)	23 (27.1)	0.361	-
	Transverse colectomy	2 (1.9)	0 (0.0)	2 (2.4)		
	Left hemicolectomy	4 (3.7)	1 (4.5)	3 (3.5)		
	Anterior resection	43 (40.2)	6 (27.3)	37 (43.5)		
	Low anterior resection	25 (23.4)	6 (27.3)	19 (22.3)		
	Hartmann’s operation	1 (0.9)	0 (0.0)	1 (1.2)		
Presence of stoma	Yes	14 (13.1)	4 (18.2)	10 (11.8)	0.431	1.667 (0.469–5.926)
	No	93 (86.9)	18 (81.8)	75 (88.2)		
Stage III risk ^2^	Low risk	67 (62.6)	14 (63.6)	53 (62.4)	0.912	0.946 (0.358–2.505)
	High risk	40 (37.4)	8 (36.4)	32 (37.6)		
Lymphatic invasion	Yes	46 (43.0)	10 (45.5)	36 (42.4)	0.793	1.134 (0.442–2.913)
	No	61 (57.0)	12 (54.5)	49 (57.6)		
Venous invasion	Yes	18 (16.8)	7 (31.8)	11 (12.9)	0.052	3.139 (1.047–9.414)
	No	89 (83.2)	15 (68.2)	74 (87.1)		
Perineural invasion	Yes	41 (38.3)	7 (31.8)	34 (40.0)	0.482	0.700 (0.258–1.896)
	No	66 (61.7)	15 (68.2)	51 (60.0)		
CEA (ng/mL)	Median (range)	9.83 (0.54–164.60)	6.69 (1.06–40.97)	10.64 (0.54–164.60)	0.429	0.794 (−5.92–13.83)
CA19-9 (U/mL)	Median (range)	85.87 (0.60–7098.0)	14.77 (0.60–54.64)	104.94 (0.60–7098.0)	0.591	0.539 (−81.77–262.12)

Abbreviations: RDI, relative dose intensity; DM, diabetes mellitus; HTN, hypertension; BMI, body mass index; CEA, carcinoembryonic antigen; CA19-9, carbohydrate antigen 19-9; OR, odds ratio; CI, confidence interval. ^1^ Tumor location: Right side: ascending colon, proximal two-thirds of the transverse colon; left side: distal one-third of the transverse colon, descending colon, sigmoid colon, rectum. ^2^ Stage III risk: low risk: T1-3 and N1, high risk: T4 and/or N2.

**Table 2 jcm-12-01423-t002:** Comparison of nutrition, inflammatory index, and body composition according to RDI.

Variable		Low RDI	High RDI	*p*-Value
		(Mean ± SD)	(Mean ± SD)	
Total protein (g/dL)		6.57 ± 0.40	6.85 ± 0.44	0.010
Serum albumin (g/dL)		3.85 ± 0.30	4.05 ± 0.34	0.005
Total cholesterol (mg/dL)		156.09 ± 39.41	175.52 ± 38.67	0.070
Hemoglobin (g/dL)		11.86 ± 1.39	11.94 ± 1.36	0.686
WBC count (10^3^/µL)		5.61 ± 1.71	6.81 ± 2.05	0.013
Pre-chemotherapy CONUT score ^1^ (N (%))	Normal (0–1)	18 (81.8)	76 (89.4)	0.462
	High (≥2)	4 (18.2)	9 (10.6)	
PNI		70.02 ± 105.73	50.15 ± 4.94	0.072
NLR		2.28 ± 2.25	2.32 ± 1.17	0.903
PLR		206.45 ± 167.47	185.63 ± 85.82	0.419
SFI (cm^2^/m^2^)		60.33 ± 23.93	58.55 ± 28.66	0.595
VFI (cm^2^/m^2^)		51.56 ± 27.44	47.65 ± 26.48	0.541
TFI (cm^2^/m^2^)		111.90 ± 45.96	106.20 ± 46.25	0.649
Visceral–subcutaneous Fat ratio		0.90 ± 0.44	0.89 ± 0.51	0.635
SMI (cm^2^/m^2^)		39.71 ± 7.62	44.63 ± 8.82	0.037
PMI (cm^2^/m^2^)		4.39 ± 0.97	5.60 ± 3.86	0.024

Abbreviations: SD, standard deviation; CONUT, controlling nutritional status; PNI, prognostic nutritional index; NLR, neutrophil-to-lymphocyte ratio; PLR, platelet-to-lymphocyte ratio; SFI, subcutaneous fat index; VFI, visceral fat index; TFI, total fat index; SMI, skeletal muscle index; PMI, psoas muscle index. ^1^ CONUT score: the sum of scores of serum albumin ^a^, total lymphocyte count ^b^, and total cholesterol ^c^. ^a^ Serum albumin (g/dL): 0 points (≥3.50), 2 points (3.00–3.49), 4 points (2.50–2.99), 6 points (<2.50). ^b^ Total lymphocyte count: 0 points (≥1600), 1 point (1200–1599), 2 points (800–1199), 3 points (<800). ^c^ Total cholesterol (mg/dL): 0 points (>180), 1 point (140–180), 2 points (100–139), 3 points (<100).

**Table 3 jcm-12-01423-t003:** Univariate and multivariate analysis according to relative dose intensity (RDI).

Variable	Univariate Analysis		Multivariate Analysis	
	OR (95% CI)	*p*-Value	OR (95% CI)	*p*-Value
Age	4.010 (1.356–11.863)	0.012	3.795 (1.152–12.495)	0.028
Sex	0.814 (0.318–2.085)	0.668		0.903
Pre-chemotherapy—BMI (1)	0.763 (0.140–4.165)	0.754		
Pre-chemotherapy—BMI (2)	0.750 (0.115–4.898)	0.764		
Tumor location	0.602 (0.228–1.587)	0.305		
Stage III risk	0.946 (0.358–2.505)	0.912		
CEA	0.985 (0.948–1.024)	0.446		
CA19-9	0.990 (0.964–1.017)	0.483		
Total protein (g/dL)	0.235 (0.075–0.730)	0.012		0.218
Serum albumin (g/dL)	0.186 (0.046–0.750)	0.018		0.537
Total cholesterol (mg/dL)	0.987 (0.974–1.000)	0.043	0.987 (0.972–1.002)	0.086
Hemoglobin (g/dL)	0.958 (0.679–1.352)	0.808		
White blood cell count (10^3^/µL)	0.649 (0.462–0.913)	0.013	0.671 (0.474–0.949)	0.024
Pre-chemotherapy CONUT score(points)	1.877 (0.519–6.783)	0.337		0.909
PNI	1.007 (0.994–1.021)	0.281		
NLR	0.979 (0.699–1.371)	0.902		
PLR	1.002 (0.998–1.006)	0.423		
SFI (cm^2^/m^2^)	1.002 (0.986–1.019)	0.787		
VFI (cm^2^/m^2^)	1.005 (0.988–1.023)	0.538		
TFI (cm^2^/m^2^)	1.003 (0.993–1.013)	0.604		
Visceral–subcutaneous Fat ratio	1.026 (0.395–2.667)	0.958		
SMI (cm^2^/m^2^)	0.921 (0.859–0.987)	0.020	0.915 (0.846–0.989)	0.025
PMI (cm^2^/m^2^)	0.635 (0.425–0.947)	0.026		0.545

Abbreviations: BMI, body mass index; CEA, carcinoembryonic antigen; CA19-9, carbohydrate antigen 19-9; CONUT, controlling nutritional status; PNI, prognostic nutritional index; NLR, neutrophil-to-lymphocyte ratio; PLR, platelet-to-lymphocyte ratio; SFI, subcutaneous fat index; VFI, visceral fat index; TFI, total fat index; SMI, skeletal muscle index; PMI, psoas muscle index.

## Data Availability

The data used in this study are available from the corresponding author upon reasonable request.

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
