# Peer review of "The Impact of Pre-Chemotherapy Body Composition and Immunonutritional Markers on Chemotherapy Adherence in Stage III Colorectal Cancer Patients"

_jcm, 2023, doi:10.3390/jcm12041423_

Round 1

Reviewer 1 Report (Previous Reviewer 3)

The authors adequately addressed my comments

Author Response

We greatly appreciate your valuable feedback. We are dedicated to continuously improve our research methodology and are determined to achieve the highest standards of excellence in this manuscript. Thank you for your support and we look forward to moving this paper closer to publication.

Reviewer 2 Report (New Reviewer)

Authors investigated body composition parameters and analyzed their effect on chemotherapy adherence in stage III CRC patients. The following questions were raised:

1. Authors retrospectively measured several body composition parameters via CT scan analysis. Apart from the retrospective design, and thinking about future studies in general, would it not be easiery, more economical and less time consuming if the body composition parameters are measured using a body compostion analyzer? Reviewer suggests that authors should also discuss this quesion.

2. Citations are needed for the followin in methods: CONUT, PNI. Similarly, citations are needed in chapter 2.4 after "According to well-known studies".

3. Lines 210-214: This is basically the same results as the previous two paragraphs, just not using t-tests or chi-squared tests but a glm...

4. Why did not authors perform survival analysis?

5. The text describing the nomogram is poor, more details is needed. In Discussion, there is basically no mention of the nomogram. Why is good to use it? Why is this nomogram better than any of the previos ones described in previous studies, etc.

6. The text needs moderate English corrections.

In summary, the topic of the article is interesting, but the presentation of the data needs additional work including survival analysis, and those that are repetative needs to be removed. Moreover, authors should also discuss all of their results, not just a selected part.

Author Response

We greatly thank the Reviewer for taking your precious time to read the manuscript carefully. We appreciate the opportunity to revise our work based on your suggestions, and are confident that the improved manuscript will be of higher quality as a result. Please find our point-by-point response to your comments in the attached file. We look forward to your further evaluation and feedback.

Round 2

Reviewer 2 Report (New Reviewer)

The article improved significantly during revison. I recommend its acceptance.

This manuscript is a resubmission of an earlier submission. The following is a list of the peer review reports and author responses from that submission.

Round 1

Reviewer 1 Report

The author used statistical analysis of the medical records of 107 patients with stage III CRC who received adjuvant Folfox chemotherapy in a single centre from 2014 to 2018 to investigate if body composition impacts chemotherapy adherence in patients with colorectal cancer. The current research paper format, in my opinion, is not well organized and is inappropriate for publication in MDPI's journal of clinical medicine. As a result, I am afraid that the manuscript might not be accepted for publication in this journal. But first, let me offer some general and particular feedback to the authors.

General comments: In order to better understand how nutrition and body composition affect chemotherapy adherence and compliance in patients with stage III colorectal cancer (CRC) who received adjuvant Folfox chemotherapy as first-line treatment, this study also looked at indicators to improve patient adherence to chemotherapy before chemotherapy was administered. The current study work style, in my opinion, is poor for publication in MDPI's journal of clinical medicine. So, I think the work could not be published in this journal.

Specific comments:       

  1. It is unfortunate to review the article without observing the line number in the body of the manuscript.
  2. Introduction: This part is very poor. The author should rewrite this part with appropriate references.
  3. The methodology and results are inconsistent.
  4. The results are presented with irrelevant and known things.
  5. The author should understand the parameters, markers, and biomarkers of cancer patients.
  6. The results are not supported by the conclusion. Based on what the main goal of the work was and what the main results were, the conclusion should be better.
  7. The language and style of English must be extensively adjusted.

Author Response

Dear, reviewer.

We thank you for the comprehensive review of our manuscript and for providing helpful suggestions. 

Please see the attachment PDF file.

Reviewer 2 Report

The authors study a relatively well-known downside of chemotherapy and how it affects patient’s adherence. Some of the findings (patient’s fat index not related to chemotherapy adherence) are quite interesting. However, some aspects need to be clarified especially regarding the type of surgery. The authors consider 121 patients diagnosed with colorectal cancer and that underwent surgery.

The authors fain to mention anything else other than disease staging (stage III) from a surgical point of view. Depending on tumor location various surgeries can be performed (left / right hemicolectomy, Dixon’s procedure, intersphincteric resection etc.) with or without a diversion ileostomy / colostomy. The type of surgery the patients was subjected to will affect the nutritional status of the patient following surgery and during FOLFOX4 regime. The presence of an ileostomy / colostomy (protective) will further affect the intake / loss of nutrients / electrolytes following surgery and this aspect will affect the BMI index, fat index and so on. The authors need to take this into account. For CRC patients, depending on the type of surgery, the authors need to consider the body mass / fatty mass loss that is due to the surgery itself and what is the percentage / amount of loss due to chemo.

Author Response

Dear, reviewer.

Please see the attachment PDF file. Thank you for your effort and attention.

Reviewer 3 Report

The manuscript from Lee and colleagues addresses associations between skeletal muscle index (SMI) and relative dose index (RDI) in colon cancer patients submitted to adjuvant treatment. A single-center retrospective study design was employed. In total, 107 patients were included. The main findings from this study suggest that low SMI predict low RDI. In general, the design and statistical analyses of the study are appropriate, and the study included multivariate analysis considering age and the variables with P< 0.1 in univariate analysis. Although the study included a relatively small number of patients, it deals with an homogeneous population of stage IIICRC in adjuvant setting and the paper is well written.

 Specific comments

 1.     Please define how RDI was calculated. Was RDI calculated for each chemotherapeutic agent?

2.     Please better describe the reasons of dose reduction or regimen discontinuation.

3.     Please describe the toxicity grades for low and high RDI.

Author Response

(The authors gave the same response as above.)

Round 2

Reviewer 1 Report

Still, the title, methodology and results are insistent. I am not happy with the reply. I am afraid that I have to reject the manuscript.